# Long-Term Treatment with Gadopentetic Acid or Gadodiamide Increases TRPC5 Expression and Decreases Adriamycin Nuclear Accumulation in Breast Cancer Cells

**DOI:** 10.3390/cells12091304

**Published:** 2023-05-03

**Authors:** Weiheng Zhang, Mengyuan Wang, Weizhen Lv, Fletcher A. White, Xingjuan Chen, Alexander G. Obukhov

**Affiliations:** 1Xi’an Key Laboratory of Stem Cell and Regenerative Medicine, Institute of Medical Research, Northwestern Polytechnical University, Xi’an 710072, China; 2Medical College, Qinghai University, Xining 810001, China; 3Stark Neurosciences Research Institute, Indiana University School of Medicine, Indianapolis, IN 46202, USA; 4Department of Anesthesia, Indiana University School of Medicine, Indianapolis, IN 46202, USA; 5Department of Anatomy, Cell Biology & Physiology, Indiana University School of Medicine, Indianapolis, IN 46202, USA

**Keywords:** TRPC5, GBCAs, breast cancer, chemotherapy resistance

## Abstract

Gadopentetic acid and gadodiamide are paramagnetic gadolinium-based contrast agents (GBCAs) that are routinely used for dynamic contrast-enhanced magnetic resonance imaging (MRI) to monitor disease progression in cancer patients. However, growing evidence indicates that repeated administration of GBCAs may lead to gadolinium (III) cation accumulation in the cortical bone tissue, skin, basal ganglia, and cerebellum, potentially leading to a subsequent slow long-term discharge of Gd^3+^. Gd^3+^ is a known activator of the TRPC5 channel that is implicated in breast cancer’s resistance to chemotherapy. Herein, we found that gadopentetic acid (Gd-DTPA, 1 mM) potentiated the inward and outward currents through TRPC5 channels, which were exogenously expressed in HEK293 cells. Gd-DTPA (1 mM) also activated the Gd^3+^-sensitive R593A mutant of TRPC5, which exhibits a reduced sensitivity to GPCR-G_q/11_-PLC dependent gating. Conversely, Gd-DTPA had no effect on TRPC5-E543Q, a Gd^3+^ insensitive TRPC5 mutant. Long-term treatment (28 days) of human breast cancer cells (MCF-7 and SK-BR-3) and adriamycin-resistant MCF-7 cells (MCF-7/ADM) with Gd-DTPA (1 mM) or gadodiamide (GDD, 1 mM) did not affect the IC_50_ values of ADM. However, treatment with Gd-DTPA or GDD significantly increased TRPC5 expression and decreased the accumulation of ADM in the nuclei of MCF-7 and SK-BR-3 cells, promoting the survival of these two breast cancer cells in the presence of ADM. The antagonist of TRPC5, AC1903 (1 μM), increased ADM nuclear accumulation induced by Gd-DTPA-treatment. These data indicate that prolonged GBCA treatment may lead to increased breast cancer cell survival owing to the upregulation of TRPC5 expression and the increased ADM resistance. We propose that while focusing on providing medical care of the best personalized quality in the clinic, excessive administration of GBCAs should be avoided in patients with metastatic breast cancer to reduce the risk of promoting breast cancer cell drug resistance.

## 1. Introduction

Gadolinium-based contrast agents (GBCAs) are clinically utilized during dynamic contrast-enhanced magnetic resonance imaging (MRI) to diagnose and monitor the growth of various tumor and cancer lesions, including metastatic breast cancers [1]. Although GBCAs are contraindicated in patients with renal failure because of a high risk of developing nephrogenic systemic fibrosis (NSF) [2,3,4], these contrast agents are mostly considered safe and are associated only with rare adverse acute effects, such as headache, dizziness, and nausea. Therefore, GBCAs are widely used for routine clinical MRIs. However, there have been alarming reports of gadolinium (3+) cation deposition in the cortical bone tissue [5], skin, liver, basal ganglia, and cerebellum [6] of patients with normal renal function following repeated administration of GBCAs [7,8,9,10]. Notably, Gd^3+^ can be detected in human tissue for years after initial exposure, and it continues accumulating there with each subsequent GBCA-enhanced MRI scan. Likely, over time, the accumulated Gd^3+^ may be slowly leaking into the interstitial fluid.

Animal studies have revealed that high doses of GBCAs may have substantial toxicity. For example, one study in a zebra fish model demonstrated that a high concentration (12.5 mM) of Gadovist, a GBCAs, can cause a significant reduction in sensory hair counts [3]. On the other hand, to date, the clinical significance of Gd^3+^ accumulation in human tissue is unclear. No association between Gd^3+^ accumulation and occurrences of side effects has been conclusively demonstrated [6,11] thus far. However, gadavist, one widely used GBCA, is reported to cause seizures in human patients if it is used inappropriately [12,13].

Remarkably, low micromolar concentrations (1–100 μM) of trivalent cations of gadolinium increase the activity of the TRPC5 channel [14,15]. TRPC5 is predominantly expressed in the nervous system [16]. Early studies have indicated that TRPC5 activation is implicated in regulating neurite outgrowth and dendritic morphogenesis [17,18] and in pilocarpine-induced seizure genesis [19]. More recent studies have demonstrated that dysregulation of TRPC5 in cancer cells is highly associated with cancer progression, especially its chemoresistance [20,21,22,23]. Ma et al. analyzed the role of TRPC5 in adriamycin-resistant human breast cancer cells (MCF-7/ADM) and provided evidence that upregulation of TRPC5 expression and activity leads to an increased expression of multidrug efflux transporter P-glycoprotein, responsible for adriamycin clearing from breast cancer cells [22]. Additionally, the authors demonstrated that TRPC5 plays a key role during extracellular vesicle (EV) formation and release from breast cancer cells. Uniquely, TRPC5 on EVs can be intercellularly transferred to non-chemoresistant breast cancer cells, conferring chemoresistance to them due to the increased activity of TRPC5 [22]. Follow-up studies revealed that increased TRPC5 activity correlates with chemoresistance to several other chemotherapeutic agents in breast and colorectal cancers, both in vitro and in vivo [23,24]. Thus, it appears that the factors upregulating TRPC5 expression may increase the risk of anticancer drug resistance in cancer patients.

Over time, Gd^3+^ deposits formed in human tissues following repeated GBCAs administrations may become depots of slowly releasing Gd^3+^, which eventually ends up in the blood plasma. Escaping Gd^3+^ may tonically potentiate the TRPC5 signaling cascade linked to conferring chemoresistance to breast cancer cells. In this study, we determined that Gd-DTPA, a major component of magnevist, a GBCA, potentiates TRPC5 activity. Additionally, we found that long-time treatment (4 weeks) of MCF-7, SK-BR-3, and MCF-7/ADM cells with Gd-DTPA and gadodiamide upregulated TRPC5 expression and decreased the accumulation of ADM in breast cancer cell nuclei, which was abolished by TRPC5 antagonist, AC1903. Our results suggest that GBCAs may increase cells’ ADM resistance, likely via targeting TRPC5 channels.

## 2. Materials and Methods

### 2.1. Cell Culture and Transfection 

HEK cells were obtained from the American Type Culture Collection and cultured in Eagle’s Minimum Essential Medium supplemented with 10% fetal bovine serum. HEK cells were transfected using the Lipofectamine 3000 reagent (Invitrogen, Carlsbad, CA, USA) in accordance with the manufacturer’s instructions. The mouse TRPC5 (NM_009428) and two TRPC5 mutants (TRPC5-E543Q and TRPC5-R593A) were used during this study. The details of mTRPC5 cloning and the introduction of mutations are described elsewhere [14,25]. The transfection mixture contained 4 µg of each channel cDNA and 0.25 µg of the histamine H1 receptor tagged with yellow fluorescent protein. The transfected cells were cultured for 24–48 h prior to electrophysiological experiments. MCF-7, SK-BR-3, and MCF-7/ADM cells were purchased from the China National Laboratory Cell Resource Sharing Service Platform. MCF-7 and MCF-7/ADM human breast cancer cells were cultured, respectively, in DMEM medium, 10% FBS, and 1% penicillin/streptomycin and DMEM medium, 10% FBS, 1% penicillin/streptomycin, 500 ng/mL ADM, and 0.01 mg/mL bovine insulin. SK-BR-3 human breast cancer cells were cultured in McCoy’s 5A medium with 10% FBS and 1% penicillin/streptomycin.

### 2.2. Patch-Clamp Electrophysiology 

TRPC5 currents were recorded with an Axopatch 200B amplifier and Digidata 1550A digitizer (Molecular Devices, San Jose, CA, USA) in whole-cell patch-clamp mode. Series resistance compensation was set to 50–70%, and the currents were filtered at 3 kHz. Acquisition control and data analyses were performed using the pCLAMP 10 software package. Cells were voltage-clamped at a holding potential of −60 mV, and the voltage ramps from −100 to +100 mV were applied in 2 s intervals. TRPC5 activity was activated either by 10 µM histamine or by a dialysis with 500 µM GTPγS added directly into the pipette solution. Experiments in which the leak current exceeded 100 pA and/or the access resistance was greater than 10 MΩ were excluded from analysis. The current densities were calculated by dividing the current amplitude values by the cell capacitance. All of the electrophysiological experiments were performed at room temperature (22–25 °C).

### 2.3. Solutions 

The standard external solution contained (in mM): 145 NaCl, 2.5 KCl, 2 CaCl_2_, 1 MgCl_2_, 10 HEPES, and 5.5 glucose (pH 7.2 adjusted with NaOH). The standard pipette solution contained (in mM): 125 CsMeSO_3_, 3.77 CaCl_2_, 2 MgCl_2_, 10 EGTA (100 nM free Ca^2+^), and 10 HEPES (pH 7.2 adjusted with Trisma base). The NMDG^+^ solution contained (in mM): 150 NMDG-Cl, 10 HEPES, and 5.5 glucose (pH 7.2 adjusted with Trisma base). The 150 NaCl solution contained (in mM): 150 NaCl, 10 HEPES, 0.5 EGTA, and 5.5 glucose (pH 7.2 adjusted with Trisma base). The osmolarity of all solutions was adjusted to 300–305 mOsm with mannitol.

### 2.4. Cell Counting Kit-8 Assay 

MCF-7, SK-BR-3, and MCF-7/ADM cells were subdivided into the control group, gadodiamide group (1 mM), Gd-DTPA group (1 mM), and GdCl_3_ group (100 µM). After 28 days of treatment, MCF-7 and MCF-7/ADM cells at the logarithmic growth stage were detached from the substrate using the trypsin treatment and then resuspended in fresh medium. Cell suspension (100 µL, 5 × 10^3^ cells) was inoculated in each well of 96-well plates and cultured overnight at 37 °C in a 5% CO_2_ incubator. The blank wells contained the plain DMEM complete medium. For the MCF-7 group, the DMEM complete medium was supplemented with various concentrations of ADM (0.01, 0.03, 0.1, 0.3, 1, 3, 10, 30, 100 µM), while, for the MCF-7/ADM group, the DMEM complete medium was supplemented with increased concentrations of ADM (0.1, 0.3, 1, 3, 10, 30, 100, 300, 1000 µM) and cultured for two days. Cells were then washed twice with PBS, and the fresh DMEM medium containing a cell counting kit-8 solution (CCK8; Selleck Chemicals, Houston, TX, USA) was added to each well. The absorbance was measured at 450 nm using a Tecan Spark instrument (Mannedorf, Switzerland).

### 2.5. Immunofluorescence 

MCF-7, SK-BR-3, and MCF-7/ADM cells were washed with PBS and fixed with 4% PFA. The primary TRPC5 (1C8) mouse monoclonal antibody (#SC-293259, Santa Cruz Biotechnology, Dallas, TX, USA, 1:100 in PBS) was added to the fixed and permeabilized cells. The cell culture plate was placed horizontally in a wet box at 4 °C and incubated overnight. After washing, a PBS-diluted Alexa Fluor 488 Donkey Anti-mouse Antibody (Thermo Fisher Scientific, Waltham, MA, USA, 1:500) or an Alexa Fluor 594 Goat Anti-Rabbit Antibody (Immunoway, Plano, TX, USA, 1:500) was added to the fixed and permeabilized cells and incubated at room temperature for 1 h, then washed with PBS. After drying, anti-fluorescence quenching sealing tablets (containing DAPI; Beyotime, Shanghai, China) were added to seal the samples. After an incubation for 10 min at room temperature, away from light, the fluorescence was observed using a confocal microscope (Olympus FluoView FV300, Tokyo, Japan).

### 2.6. Quantitative Real-Time Polymerase Chain Reaction 

Total RNA was extracted from MCF-7, SK-BR-3, and MCF-7/ADM cells lysed in the TRIzol reagent (Invitrogen) using an OMEGA kit. High-quality extracted RNA (5 µg/sample) was used for synthesis of single-strand cDNA with a TaKaRa reverse transcription kit. Quantitative real-time PCR was conducted with 45 ng of cDNA using a BIO-RAD real-time quantitative PCR instrument (CFX96). Primers used to quantify TRPC5 are listed below: the forward primer was 5′-CCCTTTCCCTGTGTGCTCATCC-3′ and the reverse primer was 5′-TGCAGAAATCCTGAGCCAAGT-3′. The GADPH primer pair was as follows: the forward primer was 5′-AACTGCTTAGCACCCCTGGC-3′ and the reverse primer was 5′-ATGACCTTGCCCACAGCCTT-3′. The forward primer for the P-glycoprotein gene was 5′-GCCGGGAGCAGTCATCTGTGGT-3′, and the reverse primer was 5′-GATCCATTCCGACCTCGCGCT-3′. The thermal cycling protocol consisted of 3 min at 94 °C, followed by 40 cycles at 94 °C for 30 s, 53 °C for 30 s, and 72 °C for 1 min. The reactions were quantified by selecting the amplification cycle when the PCR product of interest was first detected (threshold cycle, Ct). Each reaction was performed in quadruplicate, and the average Ct value was used in all analyses. To account for variability in total RNA input, the expression of each transcript was normalized to the GAPDH RNA in the samples.

### 2.7. The ADM Accumulation Assay 

Confocal microscopy (Olympus FluoView™ FV300, Tokyo, Japan) was used to observe the distribution of ADM in MCF-7, SK-BR-3, and MCF-7/ADM cells. The cells on coverslips were treated with ADM (1 µM or 10 µM for MCF-7, SK-BR-3, and MCF7/ADM cells, respectively) for 48 h, and the ADM fluorescence was observed under a confocal microscope. The excitation wavelength was 478 nm, and the emission wavelength was 596 nm. The ratios of nuclear to cytoplasmic ADM autofluorescence intensities were determined and recorded.

### 2.8. Calcium Imaging 

After 28 days of treatment with GDD (1 mM) or GdCl_3_ (100 µM), the cells were detached using trypsin and resuspended in fresh culture medium. The cell suspension was distributed among wells of a 96-well plate (100 µL, 1 × 10^4^ cells per well) and cultured in an incubator with 5% CO_2_ at 37 °C overnight. The culture medium was discarded the next day, and the cells were washed twice with HPSS. A 50 µL aliquot of Calcium 6 Assay Kit Loading Buffer was added into each well containing cultured cells, and the cells were incubated with the buffer for 1 h. The fluorescence changes were monitored using a calcium imaging reader (BioTek Cytation5, Winooski, FL, USA). Calcium 6 fluorescence was excited at 485 nm, and emitted light was collected using a 525 nm band-pass filter. 

### 2.9. Western Blot

Cells growing in the six-well plates were detached and washed 3 times with PBS pre-chilled to 4 °C. RIPA solution (Beyotime, China) supplemented with Halt protease inhibitors (cat. # 87786, Thermo Scientific, Waltham, MA, USA) was added, and ultrasonic waves were applied 3–5 times using an ultrasonic cell crusher for 3–5 s each time on ice. The supernatant was collected by centrifugation at 13,000 rpm for 15 min at 4 °C. Protein concentration was quantified using the BCA protein concentration assay kit (Bioss, Beijing, China). Protein lysates were separated using 10% SDS-polyacrylamide gel electrophoresis. The resolved proteins were transferred onto polyvinylidene difluoride membranes (PVDF), which were first incubated with the QuickBlock™ Western blocking buffer (Beyotime, China) for 1 h at room temperature. and then incubated overnight at 4 °C with either the primary TRPC5 (1C8) mouse monoclonal antibody (SC-293259, Santa Cruz Bio-technology, USA, 1:100) or the GAPDH antibody (cat. # AF0006, Beyotime, China, 1:1000). After washing the primary antibody, the secondary antibody of the corresponding species (Abbkine, USA, 1:5000), conjugated with horseradish peroxidase (HRP), was added, incubated at room temperature for 1 h to allow for specific binding, and then eluted with TBST. Equal volumes of liquid ECL A and liquid ECL B (BI, Israel) were mixed in a centrifuge tube at a 1:1 ratio while protecting the mixture from light. The resulting mixture was added to the PVDF membrane protein side to allow the chemiluminescence reaction to occur. After 1–2 min, the ECL reagent-treated PVDF membranes were transferred to the main stage of the Bio-Rad chemiluminescence imaging system (Bio-Rad, Hercules, CA, USA) for chemiluminescence signal detection. 

### 2.10. Drugs

GDD and Gd-DTPA were purchased from Selleck. Magnevist and Gadavist were purchased from Bayer Healthcare Pharmaceutical (Whippany, NJ, USA). GdCl_3_, histamine, AC1903, and other salts used in the experimental solutions were purchased from Sigma-Aldrich (St. Louis, MO, USA).

### 2.11. Statistical Methods

SigmaPlot 12.5 (Systat Software, Inc., San Jose, CA, USA) was used to perform all statistical analyses. The unpaired t-test was used to determine whether there was a statistically significant difference between two groups. The one-way ANOVA test, followed by the Student–Newman–Keuls post hoc all pair-wise multiple comparison test was used to compare the experimental groups when the data sets were normally distributed populations with equal variances. A data set was considered significantly different if the *p* value was less than 0.05. The data are presented as mean ± standard error of the mean (SEM).

## 3. Results

### 3.1. Gd-DTPA Potentiated TRPC5 Currents in HEK Cells

Figure 1A shows the structural formulas of magnevist (PubChem CID: 55466) and gadavist (PubChem CID: 86767262). They are the most clinically used GBCAs. We first employed the single-cell patch-clamp approach to determine the effects of clinical formulations of magnevist and gadavist on TRPC5 currents in HEK cells expressing TRPC5. The formulations contained all excipients, including free, unliganded Gd^3+^ chelators. Our data revealed that Gd^3+^ (100 μM) significantly potentiated the TRPC5 current (Figure 1B), and the clinical formulations of magnevist and gadavist had no effect on the TRPC5 currents induced by 1 µM histamine (Figure 1C,D). Since magnevist exhibited a trend to potentiate TRPC5 currents, we next decided to establish whether the major constituent of magnevist, Gd-DTPA, could affect TRPC5 activity. Surprisingly, we found that Gd-DTPA (1 mM) significantly potentiated histamine-induced TRPC5 currents (Figure 1E). However, its potency was lower compared to that of Gd^3+^ (100 µM, Figure 1B). Gd-DTPA also significantly potentiated TRPC5 currents elicited by the dialysis of GTPγS (500 µM) through the patch pipette (at −100 mV: from −51.8 ± 12.3 to −117.1 ± 16.9 pA/pF and at +100 mV: from 80.4 ± 9.2 to 100.4 ± 11.3 pA/pF; *n* = 14; Figure 1F,G).

After identifying that Gd-DTPA increases TRPC5 activity, we next tested the ability of Gd-DTPA to modulate histamine-activated currents through TRPC5-E543Q, a mutant of TRPC5 that can be activated downstream of the G_q/11_-PLC signaling, but is insensitive to Gd^3+^ (Figure 2A) because it lacks critical Glu-543 residue in the Gd^3+^-binding site of TRPC5 [25]. Figure 2B,C show that Gd-DTPA had no effect on the TRPC5-E543Q currents induced by either histamine or GTPγS, suggesting that the Gd-DTPA potentiating effect requires a functional Gd^3+^-binding site of TRPC5. We next investigated whether Gd-DTPA would modulate the activity of the TRPC5-R593A mutant that was reported to be less sensitive to G_q/11_-PLC activation in the absence of Gd^3+^, but could be activated downstream of G_q/11_-PLC in the presence of Gd^3+^ [14]. We determined that, consistently, Gd-DTPA also potentiated TRPC5-R593A currents (Figure 2D,E).

### 3.2. Effect of GBCAs on MCF-7, SK-BR-3, and MCF-7 /ADM Cell Viability in the Presence of ADM

Upregulation of TRPC5 activity has been recently implicated in conferring chemoresistance to breast cancer cells [22]. There was a significant disparity in TRPC5 expression levels between MCF-7 and MCF-7/ADM cells (Appendix A). Herein, we demonstrated that the major component of the contrast agent magnevist can potentiate TRPC5 currents. Therefore, we next asked whether breast cancer cells incubated with a GBCA would acquire drug-resistance. To determine this, we cultured MCF-7, SK-BR-3, and MCF-7/ADM breast cancer cells in the presence of vehicle (PBS), Gd-DTPA (1 mM), GDD (gadodiamide, 1 mM), or GdCl_3_ (0.1 mM) for 4 weeks, with the vehicle-treated MCF-7 cells serving as controls. We then performed the ADM-resistance assay, during which MCF-7 cells were treated with 0.01, 0.03, 0.1, 0.3, 1, 3, 10, 30, and 100 µM concentrations of ADM for 48 h, whereas MCF-7/ADM cells were treated with 0.1, 0.3, 1, 3, 10, 30, 100, 300, and 1000 µM concentrations of ADM for 48 h. The IC_50_ values for ADM were 0.16 ± 0.04 µM, 0.42 ± 0.17 µM, 0.75 ± 0.11 µM, and 0.17 ± 0.05 µM in GDD-, Gd-DTPA-, and GdCl_3_-treated MCF-7 cells; 1.06 ± 0.04 µM, 1.51 ± 0.05 µM, 1.21 ± 0.02 µM, and 1.12 ± 0.11 µM in GDD-, Gd-DTPA-, and GdCl_3_-treated SK-BR-3 cells; and 20.8 ± 3.1 µM, 31.9 ± 5.7 µM, 33.4 ± 9.9 µM, and 17.1 ± 3.1 µM in GDD-, Gd-DTPA-, and GdCl_3_-treated MCF-7/ADM cells, respectively. There was no significant difference in the IC_50_ values for ADM among the tested MCF-7 groups, SK-BR-3 groups, or MCF-7/ADM groups (Figure 3A–C). However, GDD-, Gd-DTPA-, and GDCl_3_-treated MCF-7 and SK-BR-3 cells exhibited increased viabilities in the presence of 1 µM ADM as compared to the control cells (Figure 3D,F), and GDD-, Gd-DTPA-, and GDCl_3_-treated MCF-7/ADM cells exhibited increased viability in the presence of 10 µM ADM compared to control MCF-7/ADM cells (Figure 3E).

### 3.3. The Nuclear Accumulation of ADM Was Decreased in the MCF-7 Cells Treated with GCBAs for 4 Weeks

To further investigate whether GCBAs can promote drug resistance in breast cancer cells, we measured the accumulation of ADM in the cytosol and nucleoplasm of MCF-7 and MCF-7/ADM cells by detecting ADM autofluorescence. As in the previous experiments, MCF-7 and MCF-7/ADM cells were first treated with GDD (1 mM), Gd-DTPA (1 mM), or GdCl_3_ (0.1 µM) for 4 weeks, and then ADM was added into the culture medium for 48 h. Figure 4A,C show that the autofluorescence of ADM decreased in the nucleus and increased in the cytoplasm of GCBA-treated MCF-7 cells compared to the control MCF-7 cells. The ratios of nuclear to cytoplasmic fluorescence intensities were 0.53 ± 0.07, 0.31 ± 0.06, 0.31 ± 0.04, and 0.15 ± 0.01 for control, GDD-, Gd-DTPA-, and GdCl_3_-treated MCF-7 cells, respectively. Conversely, the amount of ADM accumulation in the nuclei of MCF-7/ADM cells treated with GCBAs or GdCl_3_ was not different from that in the control MCF-7/ADM cells (Figure 4B,D). Notably, the ratio of nuclear to cytoplasmic fluorescence intensities was significantly smaller in MCF-7/ADM cells as compared to MCF-7 cells (0.11 ± 0.02, 0.1 ± 0.02, 0.11 ± 0.01, and 0.1 ± 0.01 for control, GDD-, Gd-DTPA-, and GdCl_3_- treated MCF-7/ADM cells).

### 3.4. The Expression of TRPC5 Was Increased after Treatment with GCBAs, and the Inhibition of TRPC5 Increased ADM Nuclear Accumulation

We next performed immunofluorescence experiments, quantitative real-time polymerase chain reaction (qRT-PCR), and Western blots to test the effect of 4 weeks of treatment with GCBAs on the expression of TRPC5 in MCF-7 cells. Figure 5A,B show that the immunofluorescence intensities were significantly greater, by 1.7-, 1.3-, and 2.1-fold, in GDD-, Gd-DTPA-, and GdCl_3_-treated MCF-7 cells probed with the anti-TRPC5 antibody as compared to the vehicle-treated MCF-7 cells. Subsequent qRT-PCR quantification and Western blotting experiments further confirmed that the expression of TRPC5 was significantly upregulated in MCF-7 cells treated with GDD, Gd-DTPA, and GdCl_3_ (Figure 5C–E). The immunofluorescence intensity of TRPC5 immunostaining was also significantly increased, by 1.2 and 1.8 times, in GDD- and GdCl_3_- treated MCF-7/ADM cells compared to the vehicle-treated MCF-7/ADM cells (Appendix A).

We assessed the accumulation of ADM in the nuclei of MCF-7 cells in the presence of AC1903, a selective inhibitor of TRPC5. MCF-7 cells were treated with Gd-DTPA for 4 weeks. We then added 1 µM ADM to measure ADM accumulation in the nuclei of MCF-7 cells. Compared to the control MCF-7 group, the ratios of nuclear to cytoplasmic ADM autofluorescence intensities were significantly decreased in the Gd-DTPA treatment MCF-7 group, and the effect was reversed by the AC1903 treatment (Figure 5F,G). Similar data were obtained in MCF-7/ADM cells (Appendix A). Thus, AC1903 increased ADM accumulation in the nuclei of MCF-7 and MCF-7/ADM cells.

### 3.5. Long-Term Treatment with GDD Decreased Intracellular Calcium Transients Induced by Histamine and Bradykinin in MCF-7 Cells

We next set out to determine whether elevated TRPC5 expression correlates with increased TRPC5 activity in MCF-7 cells treated with GDD or GdCl_3_ for 4 weeks. TRPC5 can be activated by ligands of membrane receptors coupled to G_q/11_-Phospholipase C, and MCF-7 cells express two such receptors: the histamine and bradykinin receptors. Therefore, we measured histamine- and bradykinin-elicited [Ca^2+^]_i_ changes in GDD-, GdCl_3_-, and vehicle-treated MCF-7 cells. Unexpectedly, we found that histamine- and bradykinin-induced [Ca^2+^]_i_ rises were smaller in GDD-treated MCF-7 cells compared to vehicle-treated MCF-7 cells (Figure 6).

### 3.6. The Effects of GCBA Treatment on SK-BR-3 Cells

We also determined the accumulation of ADM in the cytosol and nucleoplasm of SK-BR-3 cells by measuring ADM autofluorescence. Figure 7 shows that the autofluorescence of ADM significantly decreased in the nuclei of GCBA-treated SK-BR-3 cells compared to the control cells. The ratios of nuclear to cytoplasmic fluorescence intensities were 1.01 ± 0.14, 0.1 ± 0.03, 0.08 ± 0.03, and 0.06 ± 0.01 for control, GDD-, Gd-DTPA-, and GdCl_3_-treated SK-BR-3 cells, respectively. 

Immunofluorescence and qRT-PCR experiments were used to detect the effect of 4 weeks of treatment with GCBAs on the expression of TRPC5 in SK-BR-3 cells. Figure 8A,B show that the TRPC5 immunofluorescence intensities were significantly increased by 1.64-, 1.61-, and 1.6-fold in GDD, Gd-DTPA, and GdCl_3_-treated cells compared to the vehicle-treated cells. qRT-PCR experiments further confirmed that the expression of TRPC5 was significantly upregulated in SK-BR-3 cells treated with GDD, Gd-DTPA, and GdCl_3_ (Figure 8C). It has been reported that the ADM resistance of MCF-7/ADM is due to activation of the TRPC5-NFATc3-P-glycoprotein (P-gp) axis [26]. Therefore, we also tested the level of P-gp expression in SK-BR-3 cells using the immunofluorescence and qRT-PCR approaches. Figure 9 shows that the expression level of P-gp was significantly increased in GDD-, Gd-DTPA-, and GdCl_3_-treated SK-BR-3 cells.

## 4. Discussion

In this study, we tested the hypothesis that long-term treatment with GBCAs would affect TRPC5 expression and/or modulate breast cancer cell survival by increasing ADM resistance. Upregulated TRPC5 expression in MCF-7 cells has been linked to increased chemotherapy resistance of the cells [22,24] via the TRPC5-NFATc3-P-gp axis. We indeed found that 28 days of treatment with GDD or Gd-DTPA, two of the most clinically used GBCAs, significantly increased TRPC5 expression in MCF-7 and SK-BR-3 cells. Long-term treatment with GBCAs also reduced ADM (adriamycin, also known as doxorubicin) accumulation in the nuclei of MCF-7 and SK-BR-3 cells. ADM exhibits its anti-cancer effects in part because it can intercalate into cancer cell DNA and disrupt topoisomerase-II-dependent DNA repair [27]. Although ADM does have cytosolic anticancer effects, ADM should accumulate into the nuclei of cancer cells to slow down their proliferation; reduced nuclear ADM accumulation may lead to decreased anticancer efficacy of ADM.

TRPC5 channels have many critical physiological roles in the human body, especially the channels are highly expressed in the nervous system [28]. Recently, the role of TRPC5 in breast cancer cell resistance to chemotherapy has been discovered [22,29,30,31]. Several research groups have demonstrated that polyvalent cations, such as La^3+^ and Gd^3+^, can markedly enhance TRPC5 activity [14,15]. Gd^3+^ is not a usual constituent of the human extracellular fluids or blood plasma. However, there is a possibility that the escape of Gd^3+^ from its chelating cages of the clinically used formulations utilized for contrast-enhanced MRI would modulate TRPC5 activity in breast cancer cells to promote breast cancer cell chemoresistance. 

GBCAs are routinely used for multiple repeated dynamic contrast-enhanced MRIs to monitor disease progression in cancer patients [32,33]. Here, we found that Gd-DTPA significantly potentiated TRPC5 currents (Figure 1). However, magnevist and gadavist had no effect on the TRPC5 current. It is possible that Gd-DTPA potentiated TRPC5 currents due to the spontaneous release of Gd^3+^ from its chelating complexes. Indeed, clinical magnevist and gadavist formulations include additional unliganded chelators to reduce the harmful effect of Gd^3+^ on the human body. However, after MRI scans, the chelators are filtered by the kidneys, whereas Gd^3+^ may accumulate in several tissues. According to Bayer AG, 1 mL of magnevist contains 0.5 mmol Gd-DTPA, 5 micromole meglumine, and 1 micromole of pentetic acid, an analog of EDTA that chelates Gd^3+^. A 70 kg patient usually receives 14 mL of the solution or 7 mmol of Gd-DTPA. This means that the patient will have a maximum final concentration of about 2.5 mM of magnevist in the blood plasma during each MRI scan. We used 1 mM concentration of Gd-DTPA, which is lower than the peak concentration of GBCA in the blood plasma. Gadavist (gadobutrol) excipients also include Tris (Hydroxymethyl)aminomethane-HCl and calcobutrol sodium, which is another Gd^3+^ chelator. Interestingly, magnevist is formulated so that it is not intended to cross the blood–brain barrier, but it appears that it does, since Gd^3+^ can accumulate in the nervous tissue. Such Gd^3+^ accumulation was documented even in subjects with normal kidney function [34]. 

Our mutagenesis experiments involving the Gd^3+^ binding site of TRPC5 demonstrated that Gd-DTPA modulates TRPC5 activity via the same Gd^3+^-binding site on TRPC5 as free Gd^3+^ does (Figure 2). Considering the role of TRPC5 in breast cancer chemoresistance [26] and the repeated use of Gd-DTPA in cancer patients, we investigated the effects of long-term Gd-DTPA or GDD treatment on the expression of TRPC5 in MCF-7, SK-BR-3, and MCF-7/ADM cells. Although the treatment of Gd-DTPA or GDD did not alter the ADM IC_50_ values in breast cancer cells, Gd-DTPA or GDD significantly improved MCF-7 and SK-BR-3 cell survival in the presence of high ADM compared to the vehicle control (Figure 3). Remarkably, four weeks of treatment with Gd-DTPA or GDD significantly increased TRPC5 expression in the studied breast cancer cell lines (Figure 5 and Figure 8). 

We noticed that TRPC5 proteins were localized to the cytosol of some Gd-DTPA- or GDD-treated cells rather than to the plasma membrane, as may have been expected (Figure 5A). It was, indeed, reported that TRPC5 may accumulate in small vesicles held in reserves near the plasma membrane to be later inserted to the plasma membrane in a growth factor-PI_3_K-Rac1-PIP_5_Kα-dependent manner [35]. Such a unique cellular distribution of TRPC5 may explain our unexpected findings. Consistently, our functional test revealed that receptor-operated Ca^2+^ signaling was downregulated in GDD-treated cells (Figure 6). It is possible that the observed reduction in receptor-operated Ca^2+^ signaling, an indirect measure of TRPC5 activity in breast cancer cells, represents a negative feedback mechanism geared towards reducing chronic Gd^3+^-dependent stimulation of TRPC5 activity. The activation of TRPC5 was confirmed by the positive effect of TRPC5 inhibitor, AC1903, a treatment that increased ADM accumulation in the nuclei of MCF-7 and MCF-7/ADM cells (Figure 5 and Appendix A). Notably, upregulation of TRPC5 protein expression is crucial for P-gp induction and the development of chemoresistance in breast cancer cells [26,36]. We indeed found that the long-term treatment with GBCAs elevated the P-gp expression in SK-BR-3 cells (Figure 9).

ADM exhibits a natural red autofluorescence, and its beneficial antitumor effect depends on the drug delivery to the nucleus [37]. Altered accumulation of ADM in MCF-7 and SK-BR-3 cells following treatment with Gd-DTPA or GDD was detected by measuring its autofluorescence (Figure 4 and Figure 7). As expected, ADM mainly accumulated inside the nuclei in control cells, but was redistributed to the cytoplasm in MCF-7 and SK-BR-3 cells treated with Gd-DTPA or GDD. In untreated MCF-7/ADM, the ratio of ADM autofluorescence in the nucleus to that in the cytoplasm was already very low, which is consistent with the high expression of TRPC5 and, consequently, P-glycoprotein in the cells (Figure 4). Notably, the ratio of nuclear to cytosolic ADM autofluorescence was impacted by GBCAs to a lesser degree in MCF-7/ADM cells compared to MCF-7 cells. This is likely because TRPC5 expression was already at a relatively high level in MCF-7/ADM compared to MCF-7 cells (Appendix A), probably driving a higher P-gp expression. However, GBCA treatment was still effective in increasing MCF-7/ADM cell survival in the presence of ADM. This is possibly because the ability of cancer cells to survive chemotherapy is not only the function of AMD efflux. It is a more complex process which also involves calcium influx through TRPC5 itself, leading to a calcium-dependent acceleration of the breast cancer cell cycle [38]. Likely, MCF-7/ADM cells are more resistant to ADM treatment if treated with GBCAs, even though nuclear accumulation of ADM is not greatly impacted by GBCAs, because GBCAs can also directly increase TRPC5-mediated calcium influx. This, apparently, is sufficient to facilitate breast cancer cell cycle rate and promote cell survival. However, future in vivo studies will be required to validate our in vitro observations. 

Many studies revealed that the repeated administration of GBCAs may lead to gadolinium (III) accumulation in various human tissues. However, the association between gadolinium (III) accumulation and the occurrence of side effects and toxic damage has not been conclusively demonstrated. Our findings reveal a possible adverse effect of the excessive administration of GBCAs used for monitoring breast cancer progression. We propose that a careful assessment of benefits versus harm should be performed for each individual patient, involving the patient in the informed decision-making process, with the goal of identifying the appropriate risk versus benefit balance.

## 5. Conclusions

Treatment with Gd-DTPA or GDD significantly increased TRPC5 expression and decreased the accumulation of ADM in the nuclei of MCF-7 and SK-BR-3 cells, increasing the risk ofchemoresistance in the breast cancer cells. It may be worthwhile to minimize repeated administration of GBCAs in breast cancer patients in order to reduce the potential of promoting chemotherapy resistance in the patients.

## Figures and Tables

**Figure 1 cells-12-01304-f001:**
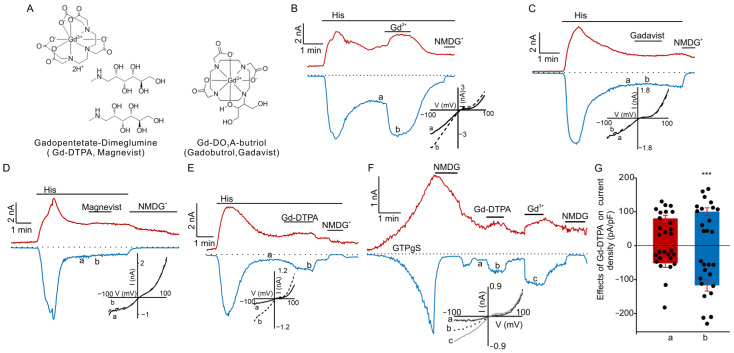
Effects of GBCAs on histamine or GTPγS-induced currents in TRPC5-expressing HEK cells. (**A**) Molecular structures of magnevist (Gd-DTPA) and gadavist. (**B**–**E**) Sample traces of TRPC5 currents induced by histamine (His, 10 µM) in the presence of Gd^3+^ (**B**), gadavist (**C**), magnevist (**D**), or Gd-DTPA (**E**). (**F**) Sample trace of TRPC5 inward (blue) and outward (red) currents, induced by dialysis of GTPγS (500 μM) via the patch pipette. Gd^3+^ (100 µM), Gd-DTPA (1 mM), magnevist (1 mM), or gadavist (1 mM) were added at the times indicated by the horizontal bars. The dotted lines indicate the zero current. The upper and lower traces represent the outward and inward whole cell currents recorded at +100 mV and −100 mV, respectively. Inserts show the current voltage relations acquired during the voltage ramps from −100 mV to +100 mV in the absence (solid lines) and presence (broken lines) of GBCAs or Gd^3+^ at the time points indicated with “a” and “b” in the same experiment. (**G**) Comparison of the mean current densities of GTPγS-activated currents, measured at the time points of “a” and “b” at holding potentials of −100 mV and +100 mV, in (**F**). *** *p* < 0.001.

**Figure 2 cells-12-01304-f002:**
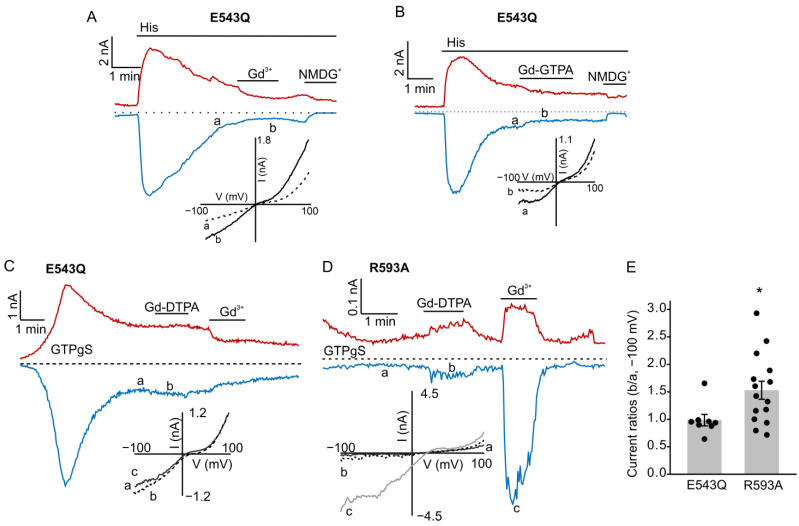
Effects of Gd-DTPA on histamine- or GTPγ-S-induced currents in TRPC5-E543Q- and TRPC5-R593A-expressing HEK cells. (**A**–**D**) Sample traces of time courses for currents via TRPC5-E543Q or TRPC5-R593A. Histamine (10 µM), Gd^3+^ (100 µM), and Gd-DTPA (1 mM) were added at the times indicated by the horizontal bars. The dotted lines indicate the level of the zero current. The upper traces represent the outward whole cell currents recorded at +100 mV, whereas the lower traces represent the inward currents recorded at −100 mV. Insets show the current–voltage relationships acquired during the voltage ramps from −100 mV to +100 mV in the absence (solid lines) and presence (broken lines) of Gd^3+^ or Gd-DTPA in the same experiment. (**E**) Comparison of the ratios of current densities measured at the time points of “a” and “b” at the holding potential of −100 mV (**C**,**D**). * *p* < 0.05.

**Figure 3 cells-12-01304-f003:**
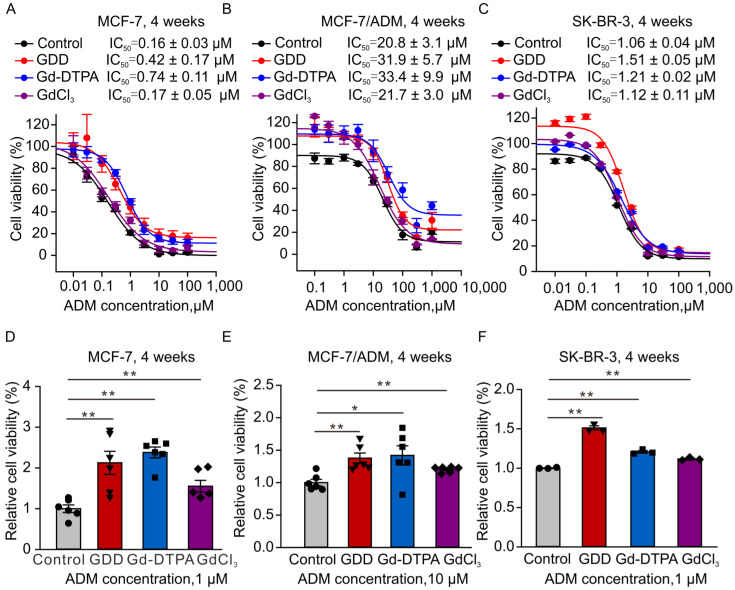
MCF-7, SK-BR-3, and MCF/ADM cell survival in the presence of ADM. (**A**–**C**) Cell viability–concentration curves were used to determine the half-maximal effective concentration (EC_50_) values for ADM in MCF-7 and MCF-7/ADM cells treated with GDD, Gd-DTPA, and GdCl_3_ for 4 weeks. (**D**–**F**) Relative cell survival of MCF-7, SK-BR-3, and MCF-7/ADM cells treated with 1 µM or 10 µM ADM for 48 h. The survival rates were analyzed using the cell counting kit-8 assay (*n* = 6). The one-way ANOVA test, followed by the Student–Newman–Keuls post hoc all pair-wise multiple comparison test, was used to compare the data of the treatment groups with those of the control group. All values are represented as means ± SEMs. * *p* < 0.05, ** *p* < 0.01.

**Figure 4 cells-12-01304-f004:**
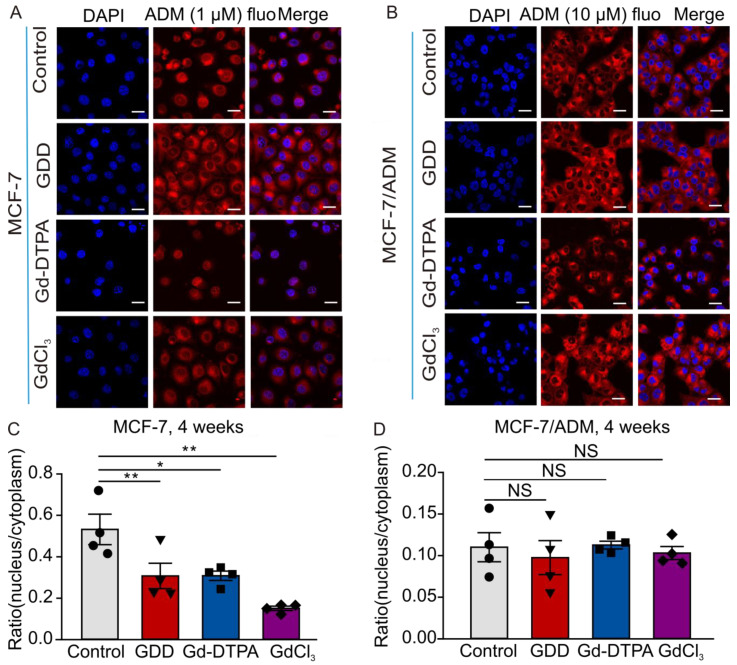
The accumulation of ADM in MCF-7 and MCF-7/ADM cells treated with GDD, Gd-DTPA, or GdCl_3_ for 4 weeks. (**A**,**B**) Confocal fluorescence images of MCF-7 and MCF-7/ADM cells (ADM autofluorescence—red; DAPI nuclear stain—blue). (**C**,**D**) Summary data of ADM accumulation in MCF-7 and MCF-7/ADM cells (*n* = 4). The one-way ANOVA test, followed by the Student–Newman–Keuls post hoc all pair-wise multiple comparison test, was used to compare the data sets to the control group. All values are represented by means ± SEMs. * *p* < 0.05, ** *p* < 0.01. NS, no significant difference. Scale bars: 30 µm.

**Figure 5 cells-12-01304-f005:**
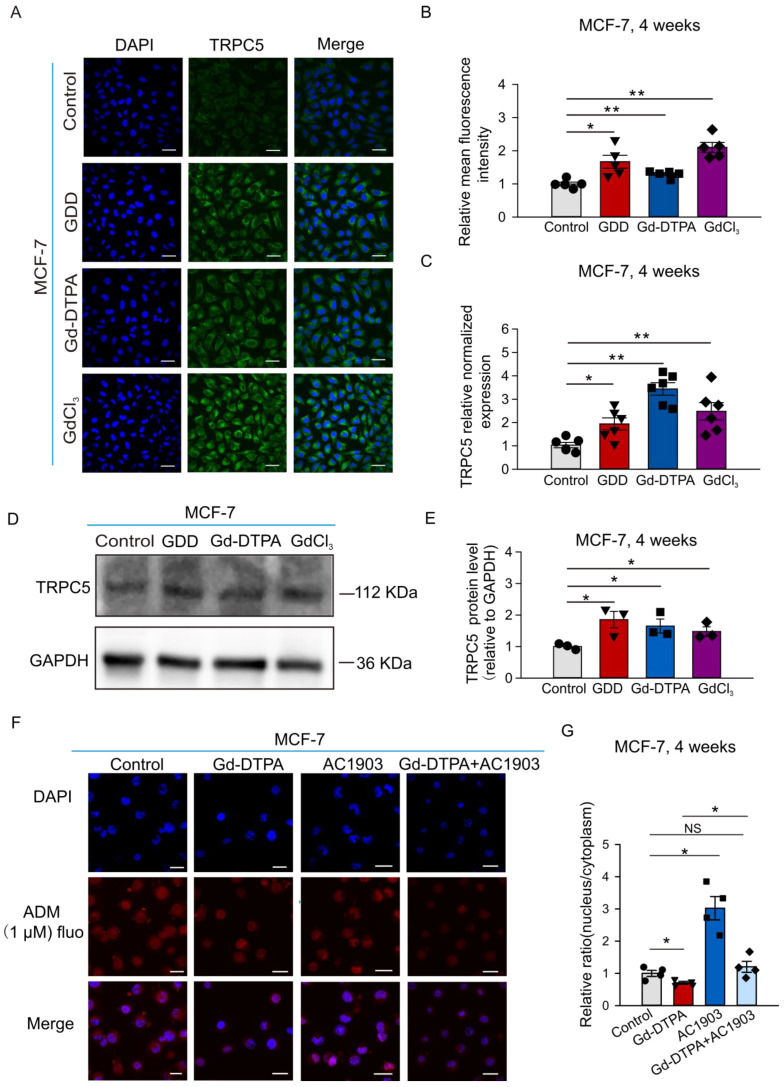
TRPC5 expression was greater in MCF-7 treated with GDD, Gd-DTPA, and GdCl_3_ for 4 weeks, and the inhibitor of TRPC5 decreased the efflux of ADM from MCF-7 cell nuclei (**A**,**B**) Representative confocal fluorescence images of TRPC5 protein immunostaining and summary data in MCF-7 cells (TRPC5, green fluorescence, *n* = 5). (**C**) qRT-PCR results for TRPC5 expression in MCF-7 cells (*n* = 5–6). (**D**,**E**) Representative Western blot images and summary data regarding TRPC5 protein expression quantification in MCF-7 cells (*n* = 3). (**F**) The nuclear ADM accumulation assay. Shown are representative confocal micrographs of MCF-7 cells treated with Gd-DTPA alone or in combination with AC1903, and in the presence of 1 µM ADM (red autofluorescence). (**G**) Summary data for nuclear ADM accumulation in MCF-7 cells (*n* = 4). The one-way ANOVA test, followed by the Student–Newman–Keuls post hoc all pair-wise multiple comparison test was used to compare the treatment groups with the control group. All values are represented by means ± SEMs. * *p* < 0.05 ** *p* < 0.01. “NS” stands for no significant difference. Scale bars: 30 µm.

**Figure 6 cells-12-01304-f006:**
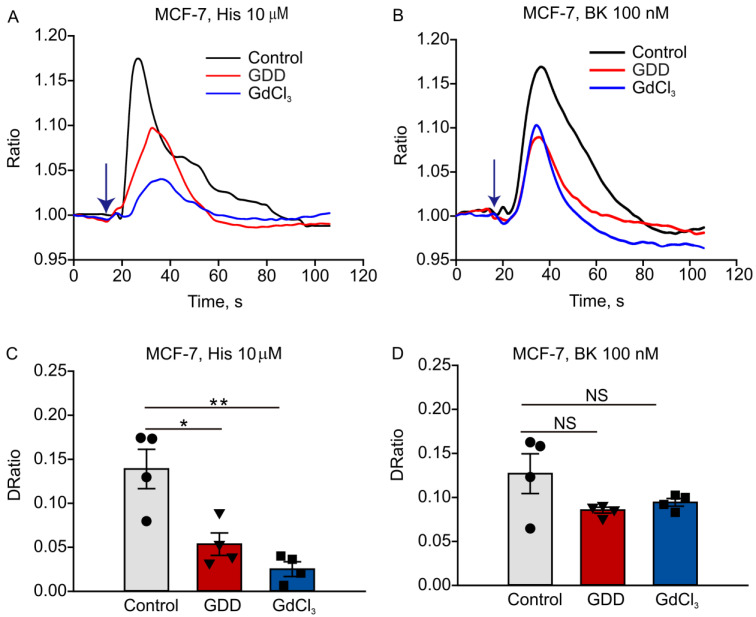
Histamine and bradykinin-induced intracellular calcium increases were smaller in MCF-7 cells receiving long-term treatment with GDD (1 mM) or GdCl_3_ (0.1 mM) compared to control MCF-7 cells. (**A**,**B**) Averaged traces show fluorescence intensity ratio changes in MCF-7 cells. The blue arrows indicate the time at which histamine or bradykinin was added to the bath. (**C**,**D**) Summary data comparing the peak fluorescence intensity ratio values shown in (**A**,**B**), *n* = 4 independent experiments. The one-way ANOVA test, followed by the Student–Newman–Keuls post hoc all pair-wise multiple comparison test, was used to compare the data sets. All values are represented by means ± SEMs. * *p* < 0.05, ** *p* < 0.01. NS, no significant difference.

**Figure 7 cells-12-01304-f007:**
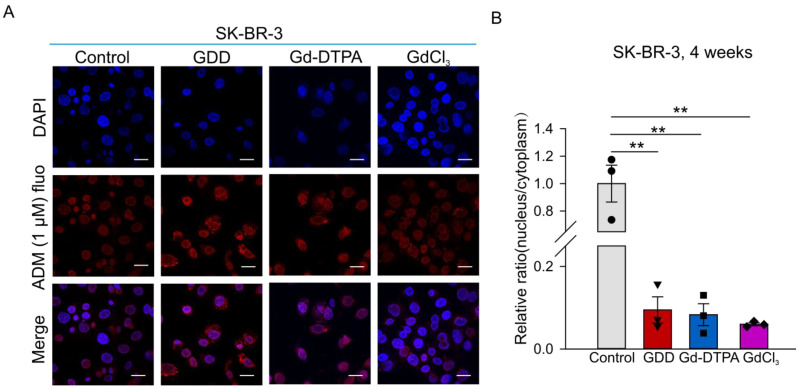
The accumulation of ADM in SK-BR-3 cells treated with GDD, Gd-DTPA, or GdCl_3_ for 4 weeks. (**A**) Confocal fluorescence images of SK-BR-3 cells (ADM autofluorescence—red; DAPI nuclear stain—blue). (**B**) Summary data of ADM accumulation in SK-BR-3 cells (*n* = 3). The one-way ANOVA test, followed by the Student–Newman–Keuls post hoc all pair-wise multiple comparison test, was used to compare the data sets to the control group. All values are represented by means ± SEMs. ** *p* < 0.01. Scale bars: 30 µm.

**Figure 8 cells-12-01304-f008:**
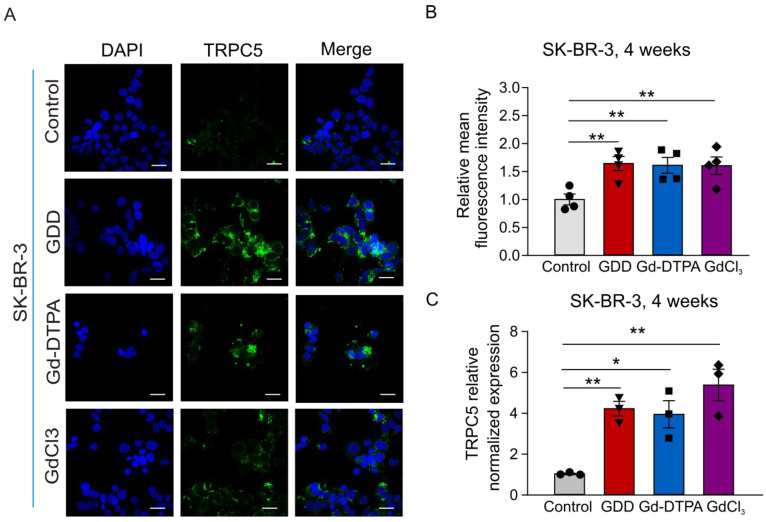
TRPC5 expression was greater in SK-BR-3 cells treated with GDD, Gd-DTPA, and GdCl_3_ for 4 weeks. (**A**,**B**) Representative confocal fluorescence images of TRPC5 protein immunostaining and summary data in SK-BR-3 cells (TRPC5, green fluorescence, *n* = 3). (**C**) qRT-PCR results for TRPC5 expression in SK-BR-3 cells (*n* = 3). The one-way ANOVA test, followed by the Student–Newman–Keuls post hoc all pair-wise multiple comparison test, was used to compare the treatment groups with the control group. All values are represented by means ± SEMs. * *p* < 0.05 ** *p* < 0.01. Scale bars: 30 µm.

**Figure 9 cells-12-01304-f009:**
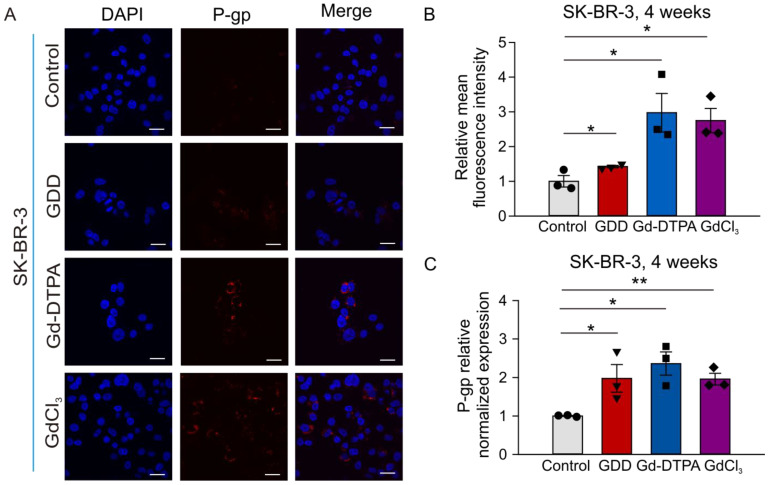
P-gp expression was greater in SK-BR-3 cells treated with GDD, Gd-DTPA, and GdCl_3_ for 4 weeks. (**A**,**B**) Representative confocal fluorescence images of P-gp protein immunostaining and summary data in SK-BR-3 cells (TRPC5, green fluorescence, *n* = 3). (**C**) qRT-PCR results for P-gp expression in SK-BR-3 cells (*n* = 3). The one-way ANOVA test, followed by the Student–Newman–Keuls post hoc all pair-wise multiple comparison test, was used to compare the treatment groups with the control group. All values are represented by means ± SEMs. * *p* < 0.05 ** *p* < 0.01. Scale bars: 30 µm.

## Data Availability

The data presented in this study are available on request from the corresponding authors.

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
