# Peer review of "Long-Term Treatment with Gadopentetic Acid or Gadodiamide Increases TRPC5 Expression and Decreases Adriamycin Nuclear Accumulation in Breast Cancer Cells"

_cells, 2023, doi:10.3390/cells12091304_

Round 1

Reviewer 1 Report

 The authors talk about Long-term treatment with gadopentetic acid or gadodiamide 2 increases TRPC5 expression and decreases adriamycin nuclear 3 accumulation in breast cancer cells and suggest that the repeated administrations should be minimized.

The following questions need to be clarified:

repeated administration of

Repeated means how many times it is being done now?

We propose that clinically, repeated 30 administration of GBCAs should be minimized in

Should be minimized to how many times?

Animal studies revealed that high doses of GBCAs may have a substantial toxicity

Need numbers/values for high doses

Need quantitative data all the time.

High dose, should be minimized, etc. are subjective terms. In Science, need objective, quantitative %s/#s/values

low micromolar concentrations-how much?

GBCAs may increase the cells ADM resistance

How much? How/why?

HEK cells were obtained fr

Expand HEK the first time.

Why HEK cells?

Nice Figures and Fig. Captions

3.1 After Fig. 1A, Figs. 1C and D are described/discussed. Fig. 1B mention/description/discussion?

No mention about Fig. 2A

3.2 very good-this is how all paragraphs must be written-with numbers and detailed descriptions of each part of Fig.

Figs. 4B and D are mentioned, what about A and C?

we tested the hypothesis whether long-term t

we tested the hypothesis that long term ---, change whether to that

It may worthwhile to minimize repeated administration

minimize to what #?

Author Response

Specific responses to Reviewer 1

Reviewer 1: “repeated administration of Repeated means how many times it is being done now?”

Response: Thank you for this important question. The repeated administrations of GBCAs are widely varying from case to case, according to the literature. For example, a clinical study described a small cohort of patients who received 28.2±5.3 doses of gadoterate meglumine, a GBCA (PMID: 28956156). Generally, the current guidelines recommend 1 to 2 preventive DCE-MRI-based breast cancer screenings per year for undiagnosed women with the identified risk factors (breast cancer associated mutations in BRCA1/2, PTEN, TP53 genes and/or the previous medical history of radiotherapy to the chest before 30-year-old), starting from the age of 30-year-old. Additionally, it depends on the referring primary care physician’s discretion. In patients with stage II or III breast cancers, the treatment protocol includes at least three cycles of neoadjuvant chemotherapy treatment prior to cancer resection surgery with four diagnostic DCE-MRIs: before chemotherapy initiation, between the chemotherapy treatment cycles, and before the cancer resection surgery (PMID: 22623692). This represents four DCE-MRIs, with additional DCE-MRIs being administered after the surgery. Chemotherapy before breast cancer resection is required to eradicate the possible metastatic foci. However, we are concerned that excessive gadolinium-enhanced MRI scans during neoadjuvant chemotherapy may promote the chemoresistance within metastatic breast cancer foci. We propose that reducing the number of DCE-MRIs may be beneficial in some patients with metastatic breast cancers.

Reviewer 1: “…repeated administration of GBCAs should be minimized in Should be minimized to how many times?

Response: We are not in the position to set the guidelines regarding how many DCE-MRIs should be performed in the clinical setting to provide the best personalized quality medical care. We believe that this question can only be addressed on a case-by-case basis. The careful assessment of benefits versus harm should be done for each individual patient with a goal to identify the risk-benefit balance, involving the patient in the decision-making process. We revised our manuscript to address this point. We now state the following “We propose that while focusing on providing the best personalized quality medical care in the clinic, excessive administration of GBCAs should be avoided in patients with metastatic breast cancers to reduce the risk of developing drug resistance” and “We propose that the careful assessment of benefits versus harm should be performed for each individual patient, involving the patient in the informed decision-making process, with a goal to identify the appropriate risk versus benefit balance.” We believe that large clinical trials will be needed to establish a general guideline for DCE-MRI frequency in breast cancer patients.

Reviewer 1: “Animal studies revealed that high doses of GBCAs may have a substantial toxicity Need numbers/values for high doses. Need quantitative data all the time. High dose, should be minimized, etc. are subjective terms. In Science, need objective, quantitative %s/#s/values low micromolar concentrations-how much?

Response: In clinic, gadopentetate dimeglumine (Magnevist) is given at a dose of 0.1 mmol/kg of body weight that corresponds to a final plasma concentration of about 2-2.5 mM (PMID: 20455064). The following reference offered some quantitative data for animal studies: Sprague-Dawley rats (4 weeks old) received intraperitoneal injections of gadoterate meglumine (0.6 or 2.5 mmol/kg), gadodiamide (0.6 or 2.5 mmol/kg) for MRI studies. These values are comparable to the GBCA doses used for DCE-MRI scans in breast cancer patients, assuming the much higher metabolic rate in rodents. This study showed that gadolinium was retained in the spinal cord and peripheral nerves in rats exposed to multiple administrations of linear and macrocyclic contrast agents. Gadodiamide (linear contrast agent) but not gadoterate meglumine (macrocyclic contrast agent) led to pain hypersensitivity, but neither affected spatial working memory performance, hippocampal cellular proliferation, or hippocampal neurogenesis (PMID: 32808889).

Reviewer 1: “GBCAs may increase the cells ADM resistance. How much? How/why?”

Response: As shown in Figure 4, GDD and Gd-DTPA decreased the ratios of nuclear to cytoplasmic of AMD from 0.53 ± 0.07 to 0.31 ± 0.06 and 0.31 ± 0.04, respectively. We propose that the treatment of GBCAs increases TRPC5 expression in MCF-7 cells and the upregulated TRPC5 promotes in turn multidrug efflux transporter P-glycoprotein expression as well as the extracellular vesicle formation and release as it was published by others (PMID: 24733904), resulting in the decreased nuclear accumulation of ADM.

Reviewer 1: “HEK cells were obtained from? Expand HEK the first time. Why HEK cells?”

Response: The Human Embryonic Kidney 293 (HEK 293) cell line is one of the cell lines commonly used in biotechnology and research. We purchased HEK cells from ATCC and have used the cells over a long time and performed many control experiments to prove that only the exogenously expressed TRPC5 protein is responsible for the observed effects of gadolinium and GBCAs (PMID: 18247362, PMID: 27920205, PMID: 15334657, PMID: 10837492, PMID: 21795434, PMID: 15689561).

Reviewer 1: “Nice Figures and Fig. Captions”

Response: Thank you for this kind comment.

Reviewer 1: “3.1 After Fig. 1A, Figs. 1C and D are described/discussed. Fig. 1B mention/description/discussion? No mention about Fig. 2A”

Response: Thank you for noting this oversight. We added the descriptions of Figure1B and Figure 2A in the revised manuscript.

Reviewer 1: “3.2 very good-this is how all paragraphs must be written-with numbers and detailed descriptions of each part of Fig. Figs. 4B and D are mentioned, what about A and C?”

Response: The descriptions of Figure4A and Figure 4C are now provided in the revised manuscript.

Reviewer 1: “we tested the hypothesis that long term ---, change whether to that”

Response: We revised this sentence as you recommended.

Reviewer 1: “It may worthwhile to minimize repeated administration minimize to what #?”

Response: Please see our clarifications above. We cannot provide an exact value in this case. This number should be determined for each individual clinical case factoring the benefit-harm balance. Large scale clinical trials will be needed to establish the clinical guideline.

Reviewer 2 Report

The study demonstrated that the clinical administration of gadolinium-based contrast agents (GBCAs) used in dynamic contrast-enhanced MRI can result in the accumulation of gadolinium (III) cations. The release of Gd3+ was found to enhance the expression of TRPC5, which may contribute to chemoresistance in breast cancer cells. The mechanism underlying this effect is believed to involve upregulation of TRPC5 channels, which reduces the accumulation of ADM in cancer cell nuclei. To further strengthen the manuscript, addressing the major/minor suggestions provided would be necessary.

Major point:

1: To address the effect of Gd-DTPA on TRPC5 currents in HEK cells, the authors should have included a control group where HEK cells were transfected with a vector that does not express TRPC5. This would allow the authors to determine whether the observed effects of Gd-DTPA on TRPC5 currents were specific to TRPC5 or were due to non-specific effects on HEK cells. A control group is an essential component of any experimental design and helps to ensure the validity and reliability of the results.

2: To confirm the findings regarding the effect of Gd3+ on TRPC5 expression in breast cancer cell lines, the authors should consider using other breast cancer cell lines to test whether the effect is specific to the cell line they used or is a more general phenomenon. Additionally, to provide a more quantitative assessment of the effect, methods such as flow cytometry or Western blotting can be used to measure TRPC5 expression levels more precisely. Therefore, it is important to confirm the findings from in vitro studies with in vivo data to better understand the biological relevance of the observed effects.

3: Is there difference of TRPC5 expression between MCF7 and MCF7/ADM? Why the GBCAs only decreased ADM accumulation in MCF7 cell nuclei but not in MCF/ADM cell.

4: the statement is not consistent with the data.

Line277: “3.3. The nuclear accumulation of ADM was increased in MCF-7 and MCF-7/ADM cells treated

with GCBAs for 4 weeks” however, the opposite fund in MCF7 cell, no change in MCF7/ADM cells.

Line 343 “3.6. Gd-DTPA-dependent up-regulation of TRPC5 expression in MCF-7 cells reduces nuclear

ADM accumulation, whereas downregulation of TRPC5 activity decreases chemoresistance in Gd-DTPA-treated MCF-7 cells”. However, the data show blocking TRPC5 did not significantly improve the survival rate of MCF-7 and MCF-7/ADM cells. (fig7A,B,C&D)

Minor point:

1: in fig1&2 , the figure legend did not describe “c”

2: line 205: fig1A? or fig1B

3: figure 5: in figure legend the number didn’t match the bar graph. (line 319 & 320)

Author Response

Specific responses to Reviewer 2

Reviewer 2: “To address the effect of Gd-DTPA on TRPC5 currents in HEK cells, the authors should have included a control group where HEK cells were transfected with a vector that does not express TRPC5. This would allow the authors to determine whether the observed effects of Gd-DTPA on TRPC5 currents were specific to TRPC5 or were due to non-specific effects on HEK cells. A control group is an essential component of any experimental design and helps to ensure the validity and reliability of the results.”

Response: Thank you for this important comment. The control experiment was performed in HEK cells expressing the GFP cDNA. Gd-DTPA did not induce any significant currents in the control GFP expressing HEK293 cells dialyzed with GTPγS. A sample trace is shown in the following figure (Figure I,See the attachment file).

Figure I. A sample whole-cell patch-clamp recording demonstrating that Gd-DTPA induced no currents in an HEK cell expressing only the GFP cDNA.

Reviewer 2: “To confirm the findings regarding the effect of Gd3+ on TRPC5 expression in breast cancer cell lines, the authors should consider using other breast cancer cell lines to test whether the effect is specific to the cell line they used or is a more general phenomenon.”

Response: There are indeed dozens of breast cancer cells (PMID: 29158785). We agree that it would be important to test the effect of Gd-DTPA on all of them. However, in this study, we focused on MCF-7 breast cancer cells because it was shown that TRPC5 upregulation causes drug resistance in this breast cancer cell line. Remarkably, the experiments for developing drug resistance are long lasting, requiring months, and cannot be completed within the limited time allocated for revising our manuscript. To address your concern, we have now assessed the expression level of TRPC5 in 4T1 cells, a mouse breast cell line, which we have available in our laboratory. We unfortunately found that TRPC5 was not significantly expressed in 4T1 cells (see Figure II,See the attachment file). Thus, we could not perform the long-time incubation experiments with Gd-DTPA in 4T1 cells.

Figure II. Western blots are shown. 4T1 cells do not express TRPC5 proteins, whereas a significant protein expression of TRPC5 is observed in MCF-7 cells. GADPH was used as the internal control.

Reviewer 2: Additionally, to provide a more quantitative assessment of the effect, methods such as flow cytometry or Western blotting can be used to measure TRPC5 expression levels more precisely.

Response: We did Western blots to assess the expression of TRPC5 in MCF-7 cells. The results are now included in Figure 5D and Figure 5E. We found that the expression of TRPC5 in MCF-7 cells was significantly up-regulated after treatment with GDD, Gd-DTPA, and GdCl3.

Reviewer 2: “it is important to confirm the findings from in vitro studies with in vivo data to better understand the biological relevance of the observed effects.”

Response: We thank Reviewer 2 for this excellent suggestion. We absolutely agree that the in vivo experiments will need to be performed in the future. We added a limitation section in the Discussion of our revised manuscript where we indicated that “the future in vivo studies will be required to validate our in vitro observations.”  

Reviewer 2: “Is there difference of TRPC5 expression between MCF7 and MCF7/ADM?”

Response: Yes, it has been reported by other laboratory that TRPC5 is up-regulated in MCF-7/ADM cells compared to MCF-7 cells (see PMID: 22988121).

Reviewer 2: “Why the GBCAs only decreased ADM accumulation in MCF7 cell nuclei but not in MCF/ADM cell.”

Response: We found that the expression of TRPC5 in MCF-7/ADM cells was significantly up-regulated by the treatment with GBCAs (Fig. S1) and the MCF-7 /ADM cell viability was increased. However, ADM accumulation in the nucleus of MCF-7/ADM was not affected in this case. This is likely because TRPC5 expression has been already at a relatively high level in MCF7/ADM compared to MCF-7 cells before GBCA treatment. Thus, the TRPC5-dependent mechanisms leading to chemoresistance were already “saturated.” Consistently, we did not observe any significant change in nuclear ADM in MCF-7/ADM cells after the addition of GBCAs.

Reviewer 2: “the statement is not consistent with the data… Line277: “3.3. The nuclear accumulation of ADM was increased in MCF-7 and MCF-7/ADM cells treated with GCBAs for 4 weeks” however, the opposite fund in MCF7 cell, no change in MCF7/ADM cells.”

Response: We have revised the manuscript to correct this error. The statement reads now “The nuclear accumulation of ADM was decreased in the MCF-7 cells treated with GCBAs for 4 weeks.”

Reviewer 2: “the statement is not consistent with the data…Line 343 “3.6. Gd-DTPA-dependent up-regulation of TRPC5 expression in MCF-7 cells reduces nuclear ADM accumulation, whereas downregulation of TRPC5 activity decreases chemoresistance in Gd-DTPA-treated MCF-7 cells”. However, the data show blocking TRPC5 did not significantly improve the survival rate of MCF-7 and MCF-7/ADM cells. (fig7A,B,C&D)

Response: KN93 is an inhibitor of CaMKII which was reported to regulate the expression of TRPC5. Here we use KN93 to test whether the inhibition of CaMKII could block the effect of Gd-DTPA increasing the expression of TRPC5 and the resistance to ADM. However, KN93 did not inhibit the increases in TRPC5 expression induced by the treatment of Gd-DTPA (Figure III, see the attachment file  which is not included in the manuscript). Therefore, KN93 did not improve the survival rate of MCF-7 (Figure7). Conversely, the antagonist of TRPC5, AC1903, significantly reduces nuclear ADM accumulation (Figure 8).

Figure III. KN93, an inhibitor of CaMKII, did not inhibit the increases in TRPC5 expression induced by the treatment of Gd-DTPA

Reviewer 2: minor points

1: in fig1&2 , the figure legend did not describe “c”

Response: Thank you for noting this error. We revised the figure legend and the description of “C” was added.

2: line 205: fig1A? or fig1B

Response: We are sorry for this mistake. It should be Figure 1B. We corrected this error in the revised manuscript.

3: figure 5: in figure legend the number didn’t match the bar graph. (line 319 & 320)

Response: We revised the manuscript to correct the error.

Round 2

Reviewer 1 Report

good work

Author Response

Thank you for your comments to improve our manuscript!

Reviewer 2 Report

"Compared with the control group, the ratios of nuclear to cytoplasmic autofluorescence intensities were significantly decreased in the  Gd-DTPA treatment group and the effect was reversed by KN93 or AC1903 treatment  (Figure 8A and 8B). Thus, AC1903 and KN93 increase ADM accumulation into the nucleus of MCF-7 cells promoting its anti-cancer activity."

the authors mentioned here that KN93 did work. So I am still confused by the results.

Response: KN93 is an inhibitor of CaMKII which was reported to regulate the

expression of TRPC5. Here we use KN93 to test whether the inhibition of CaMKII could

block the effect of Gd-DTPA increasing the expression of TRPC5 and the resistance

to ADM. However, KN93 did not inhibit the increases in TRPC5 expression induced by

the treatment of Gd-DTPA (see Figure III below which is not included in the manuscript).

Therefore, KN93 did not improve the survival rate of MCF-7 (Figure7). Conversely, the

antagonist of TRPC5, AC1903, significantly reduces nuclear ADM accumulation

(Figure 8).

Author Response

"Compared with the control group, the ratios of nuclear to cytoplasmic autofluorescence intensities were significantly decreased in the Gd-DTPA treatment group and the effect was reversed by KN93 or AC1903 treatment  (Figure 8A and 8B). Thus, AC1903 and KN93 increase ADM accumulation into the nucleus of MCF-7 cells promoting its anti-cancer activity." the authors mentioned here that KN93 did work. So I am still confused by the results.

Response: We thank the reviewer for rising this important question. KN93 is a potent inhibitor of CaMKII. We initially hypothesized that KN93 might decrease the expression rate of TRPC5 by inhibiting CaMKII. We tested this hypothesis and found that KN93 did not affect TRPC5 expression in MCF-7 cells. However, CaMKII has many diverse effects in cancer cells. CaMKII phosphorylates nearly 40 different proteins, including enzymes, ion channels, kinases, and transcription factors and plays a critical role in the regulation of proliferation, differentiation, and survival of various cancer cells. For example, KN93 suppressed NETO2-induced proliferation and melanoma metastasis (PMID: 36738427). We hypothesize that the observed increase in ADM nuclear accumulation in KN93-treated MCF7 cells is mediated by a TRPC5-independent pathway. Since many additional experiments would be needed to determine the mechanisms of KN93 anticancer effect in MCF7 cells, that appeared to be TRPC5-independent, we decided to remove the KN93 data from the revised manuscript. We plan to report the KN93 data later in a separate manuscript.